# Structural Features of DNA in tRNA Genes and Their Upstream Sequences

**DOI:** 10.3390/ijms252111758

**Published:** 2024-11-01

**Authors:** Ekaterina A. Savina, Tatiana G. Shumilina, Viktoria A. Porolo, Georgy S. Lebedev, Yury L. Orlov, Anastasia A. Anashkina, Irina A. Il’icheva

**Affiliations:** 1Engelhardt Institute of Molecular Biology, Russian Academy of Sciences, 119991 Moscow, Russia; e.savina6118@gmail.com (E.A.S.); anastasia.a.anashkina@mail.ru (A.A.A.); 2The Digital Health Center, I.M.Sechenov First Moscow State Medical University of the Russian Ministry of Health (Sechenov University), 119991 Moscow, Russia; tshumilina2000@mail.ru (T.G.S.); vporolo@mail.ru (V.A.P.); lebedev@d-health.institute (G.S.L.); orlov@d-health.institute (Y.L.O.); 3Agrarian and Technological Institute, Peoples’ Friendship University of Russia, 117198 Moscow, Russia

**Keywords:** tRNA transcription, RNA polymerase III, TBP, regulatory regions, archaea, bacteria, eukaryotes, DNA local spatial structure, ultrasonic cleavage, DNase I cleavage, DNA–protein interactions

## Abstract

RNA polymerase III (Pol III) transcribes tRNA genes using type II promoters. The internal control regions contain a Box A and a Box B, which are recognized by TFIIIC. The 5′-flanking regions of tRNA genes clearly play a role in the regulation of transcription, but consensus sequences in it have been found only in some plants and *S. pombe*; although, the TATA binding protein (TBP) is a component of the TFIIIB complex in all eukaryotes. Archaea utilize an ortholog of the TBP. The goal of this work is the detection of the positions of intragenic and extragenic promoters of Pol III, which regulate the transcription of tRNA genes in eukaryotes and archaea. For this purpose, we analyzed textual and some structural, mechanical, and physicochemical properties of the DNA in the 5′-flanking regions of tRNA genes, as well as in 30 bp at the beginning of genes and 60 bp at the end of genes in organisms possessing the TBP or its analog (eukaryotes, archaea) and organisms not possessing the TBP (bacteria). Representative tRNA gene sets of 11 organisms were taken from the GtRNAdb database. We found that the consensuses of A- and B-boxes in organisms from all three domains are identical; although, they differ in the conservativism of some positions. Their location relative to the ends of tRNA genes is also identical. In contrast, the structural and mechanical properties of DNA in the 5′-flanking regions of tRNA genes differ not only between organisms from different domains, but also between organisms from the same domain. Well-expressed TBP binding positions are found only in *S. pombe* and *A. thaliana*. We discuss possible reasons for the variability of the 5′-flanking regions of tRNA genes.

## 1. Introduction

In eukaryotes, three DNA-dependent RNA polymerases—Pol I, Pol II, and Pol III—are responsible for the synthesis of a wide variety of RNA molecules that differ in function, size, and cellular localization [1]. Two additional polymerases function in plants [2].

RNA polymerase I (Pol I) synthesizes the precursor rRNA; RNA polymerase II (Pol II) synthesizes messenger RNAs and various regulatory noncoding RNAs; and RNA polymerase III (Pol III) synthesizes small RNAs, including all tRNAs, 5SrRNAs, 7SLRNAs, U6 small nuclear RNAs [1], and mobile genetic elements known as SINEs [3]. In bacteria and archaea, the same enzyme catalyzes the synthesis of three types of RNA: mRNA, rRNA, and tRNA [4]. Archaeal RNA polymerase is very similar to the main types of RNA polymerase in eukaryotes and may be the ancestor of all three eukaryotic RNA polymerases. Archaeal and eukaryotic transcription systems are homologous. They differ structurally and functionally from the transcription machinery in bacteria [5].

Pol I, Pol II, and Pol III differ by the number of subunits and by the general transcription factors (GTFs) used. The location of regulatory elements bound by GTFs in gene promoters varies. The TATA-binding protein (TBP) is a constant component of the transcription complexes in eukaryotic organisms. Archaea use its orthologue [6]. Binding of the GTFs to gene promoters assumes the presence of the DNA binding sites. We aim to reveal specific sequence features of tRNA gene promoters based on the largest available set of data.

The initiation of the transcription of tRNA Pol III genes begins with the recognition and binding of TFIIIC to the boxes A and B, which are located inside the tRNA genes [7,8,9]. The intragenic location of the main promoters fundamentally distinguishes the process of initiation of Pol III transcription from the analogous stage of Pol II transcription.

In Pol III transcription machinery, the TBP is a part of the TFIIIB complex, which consists of three subunits: TBP, Brf1, and Bdp1 [10,11]. TFIIIB binds to TFIIIC after its binding to intragenic promoters. Only after that, the whole transcription complex is formed.

In Pol II transcription machinery, the TBP is a part of the TFIID complex, and it is a key factor that determines the assembly of the whole transcription complex [12] after binding to the TATA box.

Earlier, we provided the analysis of the sequences and structural characteristics of genomic DNA in the regions of core promoters of mRNA genes, which are transcribed by Pol II [13]. Despite the variability of the nucleotide sequences, which differ significantly between the organisms, the statistical analysis allowed us to obtain a generalized description of the core promoter’s structural organization of genes transcribed by Pol II, as for mRNA genes [13,14], so for antisense and long intergenic noncoding RNA genes [15]. In fragments located 50 bp upstream of the TSS, we found two singular regions with characteristic mechanical properties. One is located in the vicinity of the −28 bp position relative to the TSS. This is the TATA box, which is responsible for direct interaction with the TBP. The other, namely TSS, is located at positions −3/+3. The transcription start is located at position +1.

The goal of this work is the detection of the positions of the intragenic and extragenic promoters of Pol III, which regulate the transcription of tRNA genes in eukaryotes and archaea. In particular, this study aims to identify the presence of possible TBP binding sites in the 5′-flanking regions of tRNA genes in organisms of eukaryotes and archaea. We compare them with the 5′-flanking regions of tRNA genes in bacteria, as well as with Pol II promoters in eukaryotes, and identify features of the mechanical and structural properties of double-stranded DNA in the promoter regions of different genes.

## 2. Results

### 2.1. Statistical Characteristics of the Nucleotide Sequences of the tRNA Genes and Their Upstream Regions in Bacteria, Archaea, and Eukarya

We have carried out a comparison of the statistical characteristics of the texts of nucleotide sequences of the representative sets of tRNA genes of model organisms based on the analysis of the profiles of frequencies of mononucleotide occurrences at each position along the chain, complementary to the “template” chain. This chain, which is 5′→3′orientation to the TSS from the upstream region, hereinafter, we refer to as the “upper chain”. We show it as logo representation.

Figure 1A–C shows the occurrence of frequencies of mononucleotides in the vicinity of the tRNA genes start. The total length of these fragments is 80 bp. Their logo representation with an informativity of 2.0 bits is shown in Figure 2A–C. Box-A is located in these fragments. The logo representation of a 50 bp fragment located upstream of the start of the genes is shown in Figure 3A–C with an informativity of 0.4 bits. We use two logo scaling options for fragments of a 50 bp located upstream of the start of the genes because the text entropy differs significantly between nucleotide sequences before and after the start of the genes. A logo representation of 60 bp fragments for sequences aligned from the 3′ ends of tRNA genes is shown in Figure 4A–C. Box-B is located there.

#### 2.1.1. Characteristics of Nucleotide Texts

In both bacteria, the percentage of S and W nucleotides (G, C and A, T) in the 5′-flanking regions is almost identical. Among archaea, a very slight preference for W nucleotides is noticeable in *M. barkeri*. Among eukaryotes, *S. pombe* and *A. thaliana* stand out. In them, the preference for W nucleotides is evident in all 5′-flanking regions; whereas, in *D. melanogaster*, *C. elegans*, *M. musculus*, and *H. sapiens*, it appears only in very limited areas.

Logo representations with a resolution of 0.4 bits (Figure 3A–C) of the 5′-flanking regions of tRNA genes with a length of 50 bp clearly show that the nucleotide texts of bacteria differ from those of eukaryotes and archaea by an extremely low information content. Up to position −17, the informativity of the text for both bacteria does not exceed the value of 0.016. While the texts of eukaryotes and archaea, which have higher informativity, are very variable even between organisms belonging to the same domains. In both archaea (*M. barkeri* and *H. volcanii*), these sequences are completely individualized. In unicellular fungi (*S. pombe* and *S. cerevisiae*), these sequences are also distinct. *A. thaliana* is similar in many ways to *S. pombe*, while *D. melanogaster* differs markedly from both fungi and plants. The invertebrates (*C. elegans*) stand out, because they resemble neither fungi, plants, nor insects. However, in both mammals (*M. musculus* and *H. sapiens*), the sequences upstream of the genes are very similar to each other.

However—and this is very important—at the boundary between the 5′-flanking sequence and the starting position of the tRNA gene (−1/+1) in all organisms, the easily deformable Pyr/Pu dinucleotide is most often located. This resembles the situation at the boundary between the promoter region and the start of the transcription of mRNA genes transcribed by Pol II.

In all organisms without exception, the intragenic Box A is represented as the TRGYnnARBGG consensus located at positions +8/+18. The degree of conservatism of most positions varies depending on the domain and organism.

We can allocate the main features of the consensus of Box A of archaea: the absolute conservation of position +8 (T), +10 (G), +14 (A), and +15 (G) and the high extent of the entropy component of the last three positions (16, 17, and 18). It distinguishes the consensus of Box A from archaea from the bacteria and eukaryotes. One can assume that the last three positions of the Box A in archaea do not significantly affect its function.

Between the Box A in bacteria and eukaryotes, there are also some differences. Bacteria positions +8 and +14 are 100% occupied by T and A, respectively. In eukaryotes, the percent of occupation of all but one positions varies. The only position in all eukaryotes with 100% occupation by nucleotide G is +18. Although, in the Box A of *S. pombe*, *A. thaliana*, and *D. melanogaster*, the occupancies of positions +8 and +14 are also 100%. They are nucleotides T and A, respectively. Thus, there is a different degree of conservation in all but one position, namely, position +18 in the Box A in eukaryotic organisms.

The 5′ boundary of the Box A in all organisms contains the difficult-to-deform Pu/Pyr dinucleotide, more often G/T (Figure 3A–C). Conformational movements at its boundary are, therefore, negligible. This may be important for their recognition by TFIIIC.

The D-loop of mature tRNAs is partly formed from nucleotides of the Box A sequence. This imposes certain restrictions on the possibilities of evolutionary changes in these positions. These changes must satisfy two conditions at once: they must be well recognizable for TFIIIC and possess thymines in the positions, which are mostly suitable for their transformation into the dihydrouridines of mature tRNA.

The tRNA gene samples for each organism contain genes of different lengths. Therefore, to identify the consensus of the Box B, it is necessary to align the nucleotide sequences of the tRNA genes relative to their 3′ ends. The logo representations in Figure 4A–C are the result of such an alignment. It turned out that the Box B consensus, GGTTCRAnYCY, is identical in bacteria, archaea, and eukaryotes and is the same as that known earlier for human tRNA genes and SINEs [16,17]. In archaea and eukaryotes, the first nucleotide of the B-box is located at a distance of −21 nucleotides from the 3′ end of the gene.

The same consensus of Box B in bacteria is shifted three positions towards the start of the genes. The dinucleotide step at its boundary can be anything.

The boundary position between the first nucleotide of B-box and the sequence preceding it is always the difficult-to-deform dinucleotide G/G (Figure 4A–C). Thus, as for A-box, conformational movements at its boundary are rather insignificant. We suggest this is important for their recognition by the TFIIIC.

The TψC stem–loop–stem fragment in mature tRNA is partly formed from Box B sequence. Therefore, everything said above about the limitations on evolutionary changes in the Box A sequences is also applicable to the changes in the Box B sequences.

#### 2.1.2. Characterization of Spatial Structure and Physical Properties

We analyzed the variability of the local spatial structure of nucleotide sequences, as well as its mechanical and physicochemical properties in fragments with a total length of 80 bp (50 bp of 5′-flanking regions and adjacent to them 30 bp of tRNA genes). For this purpose, profiles with a number of indices characterizing structural and physical properties were constructed. Two types of structural data were used.

We estimated the structural anisotropy along the naked DNA sequence from the profiles of changes in the stacking energy, roll, slide, and double-helix stiffness values as they change, as well as from data on the mobility of the structure when bending toward the major groove. The values of these parameters for ten possible double-stranded dinucleotides are given in the database DiProDB, http://diprodb.fli-leibniz.de (accessed on 24 April 2024) [18]. The profiles of these characteristics of double-stranded nucleotide sequences ranging from −50 to +30 bp are presented for *S. pombe* and *S. cerevisiae* in Figure 5a–f and for *A. thaliana*, *D. melanogaster*, *C. elegans*, *M. musculus*, and *H. sapiens* in Figure 6a–f. The profiles of other organisms are in the Appendix A, namely, for *E. coli* and *B. subtilis* in Appendix A and for *M. barkeri* and *H. volcanii* in Appendix A.

In order to compare the structural characteristics of the 5′-flanking regions of tRNA genes and mRNA genes, we used Figure 7a–f from our previous work [14].

In order to determine the differences in the physical characteristics of complementary chains, as well as to track changes in the width of the DNA minor groove, we used the profiles of intensity of ultrasonic cleavage as well as DNAase I cleavage. We have previously discussed in detail the relationship between these experimental parameters and the structural features of nucleotide sequences in double-stranded DNA [13,14,15,19]. Statistically reliable relative ultrasonic cleavage intensities for di- and tetranucleotides were obtained in [19]. The cleavage intensities of hexanucleotides by DNAase I were obtained in [20].

Profiles of these physical parameters for tRNA gene fragments between −50 and +30 bp are presented for *S. pombe* in Figure 8a–d and for *A. thaliana* in Figure 9a–d. Profiles for other organisms are presented in the Appendix A for *E. coli*, Appendix A for *B. subtilis*, Appendix A for *M. barkeri*, Appendix A for *H. volcanii*, Appendix A for *S. cerevisiae*, Appendix A for *D. melanogaster*, Appendix A for *C. elegans*, Appendix A for *M. musculus*, and Appendix A for *H. sapiens*.

The cleavage intensity difference profiles of complementary strands are also shown in the Appendix A for *E. coli* and *B. subtilis*, Appendix A for *M. barkeri* and *H. volcanii*, Appendix A for *S. pombe* and *S. cerevisiae*, and Appendix A for *A. thaliana*, *D. melanogaster*, *C. elegans*, *M. musculus*, and *H. sapiens*).

These data considered together provide unique structural information similar to fingerprints. Let us analyze them all in complex.

Profiles of all structural parameters in all organisms show that the structural features of DNA in the 5′-flanking regions and intragenic regions of tRNA genes differ significantly (Figure 5a–f, Figure 6a–f, Appendix A). The changes begin with the TSS region. We have previously observed similar changes in the TSS region on mRNA gene profiles [13,14,15].

However, the similarity with mRNA profiles is limited to the TSS site. There is no similarity between tRNA and mRNA profiles at the 5′-flanking regions or at intragenic regions.

Large amplitudes of changes in the intragenic regions of tRNA genes correspond to the location of base pairs in motifs and are especially noticeable in the A-box region (positions +8 to +18), since its sequence is highly conserved. However, beyond the A-box, the sequences are also not random, as they must form a cloverleaf shape in the mature tRNA. Therefore, profiles of tRNA intragenic fragments are characterized by marked fluctuations that are absent in mRNA intragenic fragments (compare Figure 5a–f, Figure 6a–f, Appendix A with Figure 7a–f).

The 5′-flanking regions of tRNA genes in all organisms from all domains also differ from the promoter regions of mRNA genes. The slide value in this region in all organisms is lower than that of the random sequence. The negative slide is the characteristic of an A-DNA structure. Therefore, naked DNA in 5′-flanking regions of the genes may carry out the B→A transition of DNA with high probability.

In bacteria, this property of 5′-flanking regions of tRNA genes may increase the affinity for some proteins. The positions of transcription factors in some promoters of *E. coli* were analyzed earlier [21,22]. They are discussed in a recently published review [23]. We have found that the enhancement of DNA mobility to bend towards major groove in Gram-positive *E. coli* starts from around position −30 bp in upstream regions of tRNA genes, while in Gram-negative *B. subtilis*, it is closer to the gene start (Appendix A). However, a discussion of possible recognition positions by bacterial regulatory proteins is beyond the scope of this work.

The structural characteristics of the 5′-flanking regions of eukaryotes show great variability (Figure 5a–f and Figure 6a–f). Moreover, unicellular fungi differ from each other. Thus, *S. pombe* shows a pronounced minimum in the roll parameter and the enhancement of mobility to bend toward the major groove at positions −36 to −28. Therefore, we believe that this is the most likely position for TBP binding. Weaker binding at −22 is also possible (Figure 5a–f). In contrast, we did not find any positions in the 5′-flanking regions of *S. cerevisiae* tRNA genes with the structural parameters required for TBP binding. This is in agreement with the experimental data obtained in [24]. Nevertheless, increased mobility in the direction of the major groove just before the TSS allows for some weak interactions there with the TBP (Figure 5a–f).

*A. thaliana* stands out among multicellular eukaryotes. Logo profiles of 5-flanking sequences of this plant (Figure 3) reveal several AT-rich motifs. The fragment at position −34 to −28 bp has the highest information content. At the same position, the structural parameters roll, slide, and “mobility to bend toward major groove” indicate the presence of a TBP binding site (Figure 6a–f), while weaker binding is also possible at −22 bp and −15 bp. Logo profiles of other multicellular eukaryotes also have at least two AT-rich motifs, but their information content is much lower than in *A. thaliana*. The structural parameter “mobility to bend toward major groove” indicates the possibility of TBR binding in the vicinity of positions −22 and −15, but with low affinity (Figure 6a–f).

We have not found an exact position for the TBP binding site in 5′-flanking regions of the genes in both archaea (Appendix A). In the profiles of *M. barkeri*, “mobility to bend toward major groove” is constantly high in the whole region (−50/−1). In *H. volcanii*, there are several well-defined maxima at the fragment −50/−27 and around −16 and −6.

The intensity profiles of ultrasonic cleavage and cleavage by DNAase I are plotted for both complementary DNA strands separately. Figure 8a–d and Figure 9a–d present such profiles for *S. pombe* and *A. thaliana*, accordingly. The profiles of the other organisms are given in the Appendix A.

The profiles of ultrasonic cleavage intensity and DNAase I cleavage intensity of DNA not only confirm the fact that the structural characteristics of the 5′-flanking regions differ from those of the tRNA gene regions but also reveal the specificity of the fragment immediately upstream of the TSS from position 18 ± 1 to transcription initiation. The cleavage intensities on this fragment of about 20 bp in length are quite low and remain nearly constant (Figure 8a–d and Figure 9a–d and Appendix A). We hypothesize that this fragment, which is 20 bp long, is in contact with TFIII3C and, possibly, with TFIIID simultaneously, thereby determining the position of their interaction.

Another surprising feature of the cleavage profiles is their inversion between complementary chains when crossing gene start. The intensity of ultrasonic cleavage reflects the conformational mobility of nucleotide sequences. We observe an inversion of the cleavage intensity when crossing gene start from low to high values of the coding strand and from high to low values of the complementary strand.

We hypothesize that such profile inversion occurs due to the presence of two types of promoters around the gene start. The intragenic promoter (Box A) provides TFIIIC complex binding, and the promoter in the 5′-flanking regions provide the TBP in complex TFIIIB recognition. Conformational movements have to be slower in that strand, which will be bound by protein. In mRNA core promoters, the TATAbox region of mRNA genes has the lowest mobility [13,14,15].

The efficiency and strength of the interaction of the DNA double helix with the TBP depends mainly on the width of the minor groove [25]. In the profiles of DNase I cleavage intensity in all organisms (including *S. pombe* and *A. thaliana*) in the 5′-flanking regions of tRNA genes, we register only very short regions where this parameter increases, but we did not find a single octanucleotide with a wide minor groove. This may be an indication of a lack of strong TBP binding.

## 3. Discussion

The dependence of tRNA transcription on 5′-flanking sequences has been discussed since 1982 [26,27]. The authors considered this genome fragment as a tool for fine-tuning the control of transcriptional activity.

Even a single nucleotide change in the promoter region can affect transcriptional activity. Such an influence was found in the transcription of SINEs [17]. We have found that nucleotide motives similar to the TATAWAAR consensus of the TATA box in mRNA promoters are also present in the 5′-flanking regions of the tRNA genes in all eukaryotic organisms. However, the informational content of these motives is only greater than 0.4 bits in two of them—*S. pombe* and *A. thaliana*.

We have studied in detail the sequences and structural characteristics of tRNA genes in organisms from three domains—organisms possessing the TBP or its ortholog (eukaryotes, archaea) and organisms lacking the TBP (bacteria). Moreover, eukaryotic organisms at different evolutionary stages of development—from unicellular to multicellular—were used.

The analysis of the nucleotide texts of tRNA genes shows that intragenic Box A is represented as the consensus TRGYnnARBGG disposed in the positions +8/+18, while intragenic B-box is represented as the consensus GGTTCRAnYCY disposed from the position N-21 from the genes ends (N) in organisms of all three domains. They are absolutely the same in all the organisms studied, but they differ in terms of the conservation of some positions. Their position in gene sequences reveal that Box A is a part of the D-loop of mature tRNA, while Box B is a part of the TψC stem–loop–stem structure. These sequences are highly conserved in tRNAs, and their role is very important, as they fix the 3D structure of mature tRNA. It seems that their evolution is extremely slow.

The analysis of the nucleotide texts of 5′-flanking regions of tRNA genes shows that 5′-flanking sequences reveal several AT-rich motifs in all organisms (Figure 3). These motifs differ in their informational content. Fragments at position −34 to −28 bp have the highest information content among all organisms only in *S. pombe* and *A. thaliana*. We have not found any motives with high information content in the other eukaryotic organisms studied. Structural parameters also indicate that only *S. pombe* and *A. thaliana* possess a site capable of providing sufficiently strong binding to the TBP. Nevertheless, these organisms also do not have a single position with an enlarged major groove.

It is important to note that all organisms have at least two positions in 5-flanking regions, where the parameter “mobility to bend toward major groove” reaches local maxima. We believe that these are the positions where TBP fixation occurs as part of the TFIIIB complex.

It is important to note two recent works that analyze the mechanism of Pol III transcription. Cryoelectron microscopy and single-molecule FRET experiments were used to elucidate a mechanism of transcription 5S rRNA genes in *S. cerevisiae* [28]. This work provides insight into the multistep process of preparing the external promoter for TFIIIB loading. It shows that a complex of the two factors TFIIIA and TFIIIC directs the TBP to its DNA binding site.

The role of chromatin rearrangements in the regulation of POL3 was discussed in [29]. The authors conclude that the mechanisms of the regulation of POL3 transcription include some external factors and that the gene sequence alone does not explain all the observed effects of regulation.

## 4. Materials and Methods

We used nucleotide sequences of tRNA genes and their 50 bp upstream regions from the genomes of model organisms presenting bacteria, archaea, and eukaria.

We analyzed high confidence sets of tRNA genes from a collection of the Genomic tRNA Database (GtRNAdb) (http://gtrnadb.ucsc.edu (accessed on 1 February 2024)) [30]. This database contains tRNA gene predictions made by tRNAscan-SE on complete or nearly complete genomes. This tool “assesses the predictions with a combination of domain-specific, isotype-specific, and secondary structure scores in two filtering stages on top of the pseudogene classification, and determines the ‘high confidence’ set of genes that are most likely to be functioned in the translation process”. [31].

The 50 bp upstream regions of each gene were taken from the NCBI GenBank using the coordinates indicated in the GtRNAdb database. To do this, we counted the positions of each nucleotide in the genomic sequence using Python code. To obtain the upstream region of complementary sequences, we considered only a 5′-3′ chain in the genomic sequence.

Bacteria were represented by *E. coli* str. K-12 substr. MG1655 (89 genes) and *B. subtilis* subsp. *subtilis* str. 168 (86 genes); archaea were represented by *M. barkeri str. Fusaro* (63 genes) and *H. volcanii* DS2 (53 genes); eukaria were represented by *S. pombe* 972h- (171 genes), *S. cerevisiae* S288c (273 genes), *A. thaliana* (TAIR 10 release) (580 genes), *D. melanogaster* (BDGP Rel. 6/dm6) (290 genes), *C. elegans* (WBceI235/ce) 11 (568 genes), *M. musculus* (GRCm39/mm39) (400 genes), and *H. sapiens* (GRCh38/hg38) (429 genes).

Logo representations were made by WebLogo (http://weblogo.threeplusone.com, accessed on 4 May 2024).

The plots of DNAase cleavage, ultrasonic cleavage, structural parameters, frequencies of occurrence, and log representation were constructed as described in our previous articles [13,14,15].

## 5. Conclusions

This work has once again shown that in order to analyze the operation of regulatory regions of genomes, it is important to use not only the textual characteristics of nucleotide sequences, but also their mechanical and structural properties.

Previous analysis of eukaryotic mRNA promoters [13,14] and promoters of antisense and long noncoding intergenic RNAs [15] transcribed by Pol II revealed the commonality of their structural and physical characteristics and also the positions of TBP binding sites.

In the present work, we have analyzed the peculiarities of the organization of tRNA genes and their 5′-flanking regions.

We have found that two undecanucleotides, namely Box A and Box B, located intragenic quite identically with respect to the beginning and the end of genes in eukaryotes and archaea. In bacteria, Box B is shifted two bp positions to the center. Their consensus sequences are identical in all the organisms of all three domains. The differences concern only the information content of a number of positions. The constancy of the location of these boxes indicates that they participate in the formation of the D-loop and TψC loop of mature tRNAs.

The 5′-flanking regions of tRNA genes of eukaryotes and archaea differ markedly from the analogous regions of eukaryotic mRNAs. They do not contain a singular region near the −28 bp position, which is the TBP receptor in mRNA promoters. In contrast to Pol II promoters, the 5′-flanking regions of tRNA genes contain several fragments capable of anchoring the TBP, but their affinity to the TBP is significantly lower than that of the TATA box mRNA. Therefore, the TBP is not a determinant regulator of transcription.

We identified specific properties in a fragment located immediately upstream of the TSS from position 18 ± 1 before transcription initiation. We hypothesize that this 20 bp fragment contacts TFIIIC and, possibly, TFIIID simultaneously, thus determining the position of their interaction.

The structural properties of bacterial DNA in the 5′-flanking regions of tRNA genes evidenced that they may interact with proteins that have an affinity for the minor groove of DNA.

## Figures and Tables

**Figure 1 ijms-25-11758-f001:**
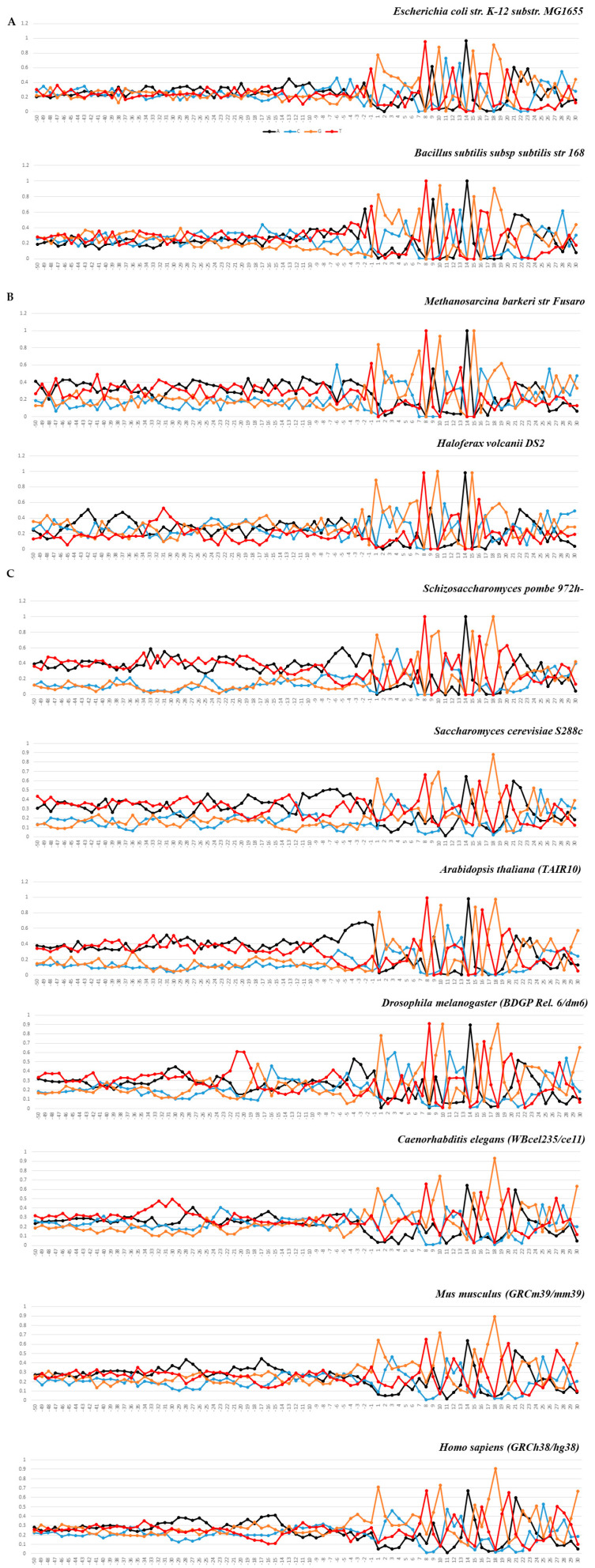
The profiles of the 50 bp upstream regions and the first 30 bp sequences of tRNA genes as the frequencies of mononucleotide occurrences (in percentages) at each position along the strand, complementary to template, for datasets of: (**A**) *E. coli* str. K-12 substr. MG1655 and *B. subtilis* subsp. subtilis str. 168; (**B**) *M. barkeri* str. Fusaro and *H. volcanii* DS2; (**C**) *S. pombe* 972h-, *S. cerevisiae* S288c, *A. thaliana* TAIR 10, *D. melanogaster* BDGP Rel. 6/dm6, *C. elegans* WBceI235/ce 11, *M. musculus* GRCm39/mm39, and *H. sapiens* GRCh38/hg38. The transcription start position is indicated by “+1” on the abscissa axis.

**Figure 2 ijms-25-11758-f002:**
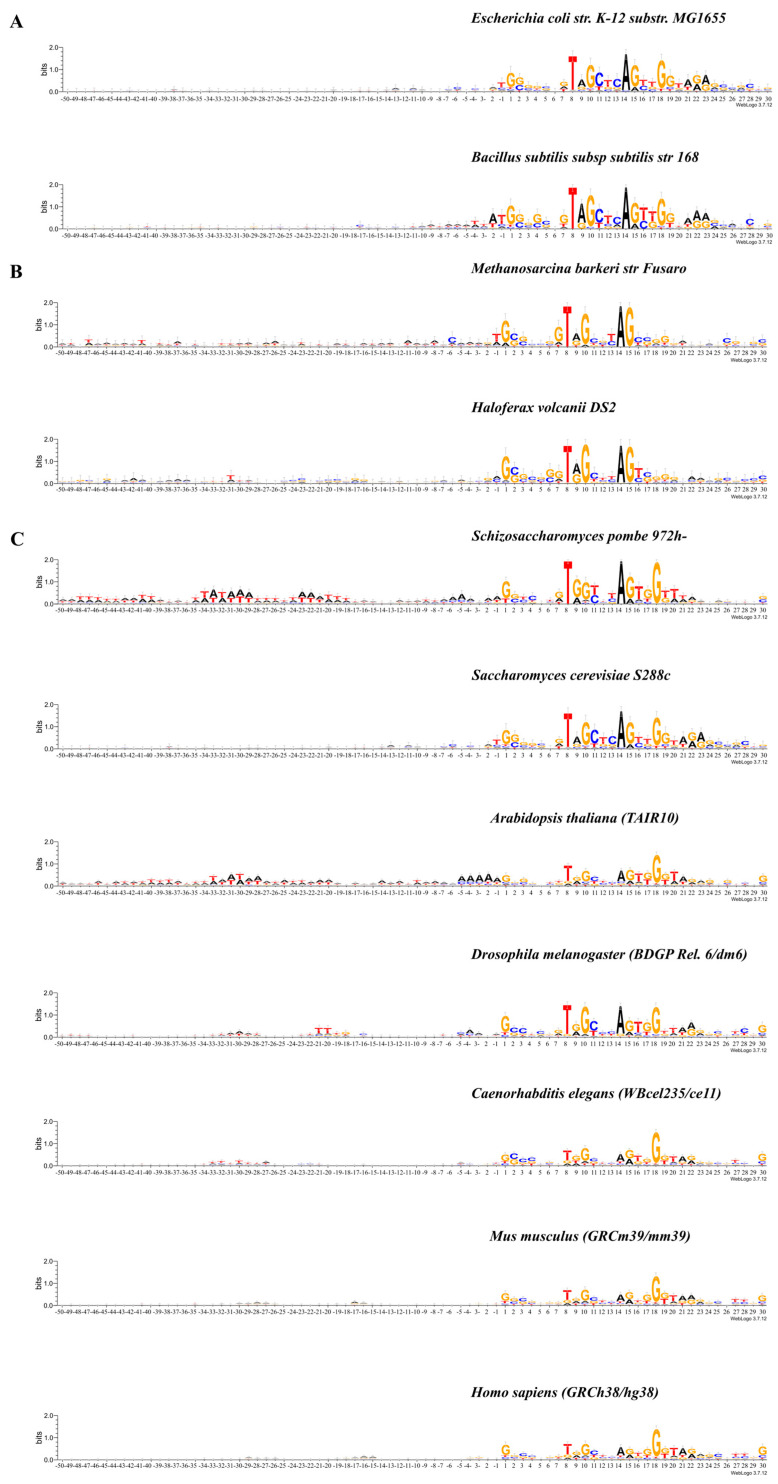
Logo representation with information content 2.0 bits of the 50 bp upstream sequences and first 30 bp fragments of tRNA genes of (**A**) *E. coli* str. K-12 substr. MG1655 and *B. subtilis* subsp. subtilis str. 168; (**B**) *M. barkeri* str. Fusaro and *H. volcanii* DS2; (**C**) *S. pombe* 972h-, *S. cerevisiae* S288c, *A. thaliana* TAIR 10, *D. melanogaster* BDGP Rel. 6/dm6, *C. elegans* WBceI235/ce 11, *M. musculus* GRCm39/mm39, and *H. sapiens* GRCh38/hg38. The transcription start position is indicated by “+1” on the abscissa axis.

**Figure 3 ijms-25-11758-f003:**
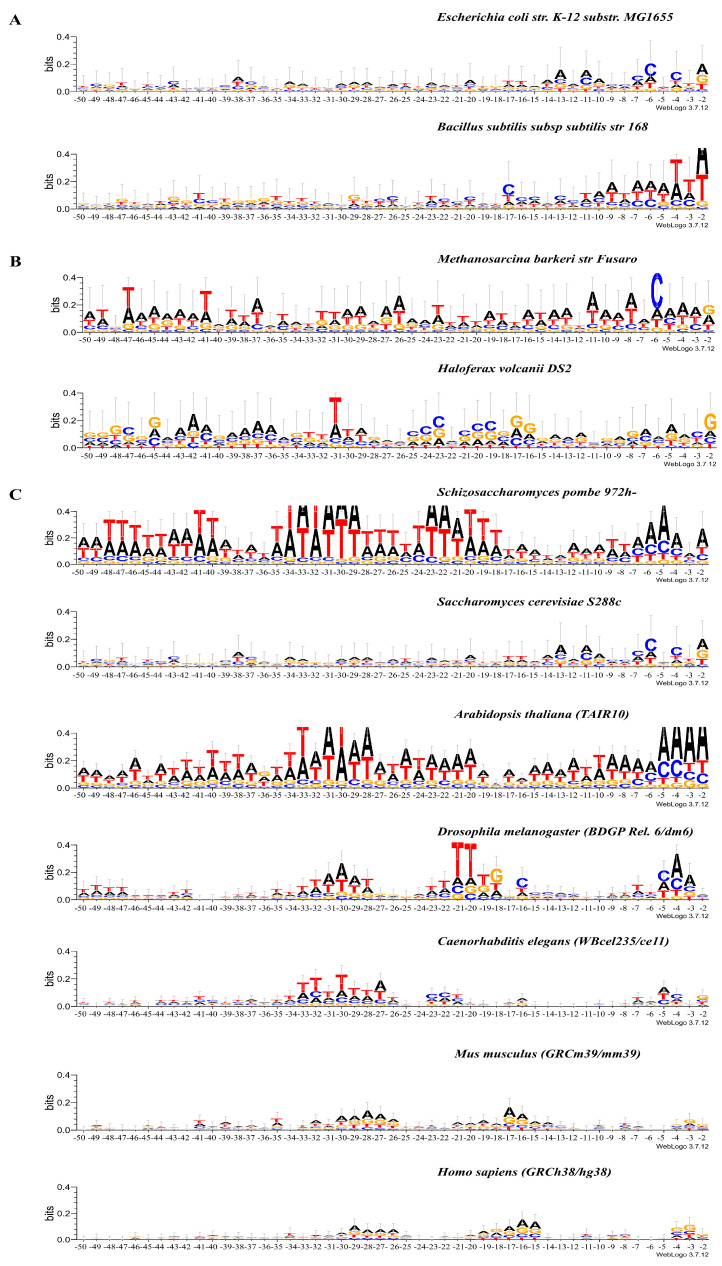
Logo representation with information content 0.4 bits of the 50 bp upstream sequences of tRNA genes of (**A**) *E. coli* str. K-12 substr. MG1655 and *B. subtilis* subsp. subtilis str. 168; (**B**) *M. barkeri* str. Fusaro and *H. volcanii* DS2; (**C**) *S. pombe* 972h-, *S. cerevisiae* S288c, *A. thaliana* TAIR 10, *D. melanogaster* BDGP Rel. 6/dm6, *C. elegans* WBceI235/ce 11, *M. musculus* GRCm39/mm39, and *H. sapiens* GRCh38/hg38.

**Figure 4 ijms-25-11758-f004:**
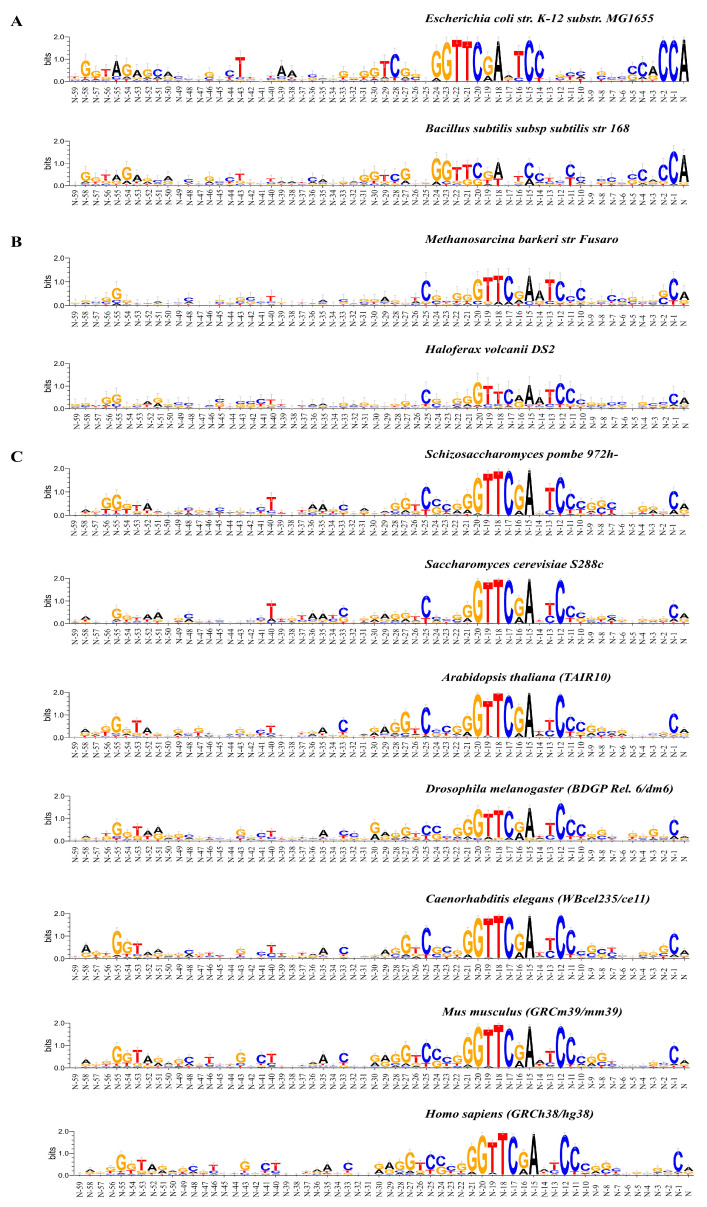
Logo representation with information content 2.0 bits of the 60 bp sequences at the 3′ ends of tRNA genes of (**A**) *E. coli* str. K-12 substr. MG1655 and *B. subtilis* subsp. subtilis str. 168; (**B**) *M. barkeri* str. Fusaro and *H. volcanii* DS2; (**C**) *S. pombe* 972h-, *S. cerevisiae* S288c, *A. thaliana* TAIR 10, *D. melanogaster* BDGP Rel. 6/dm6, *C. elegans* WBceI235/ce 11, *M. musculus* GRCm39/mm39, and *H. sapiens* GRCh38/hg38.

**Figure 5 ijms-25-11758-f005:**
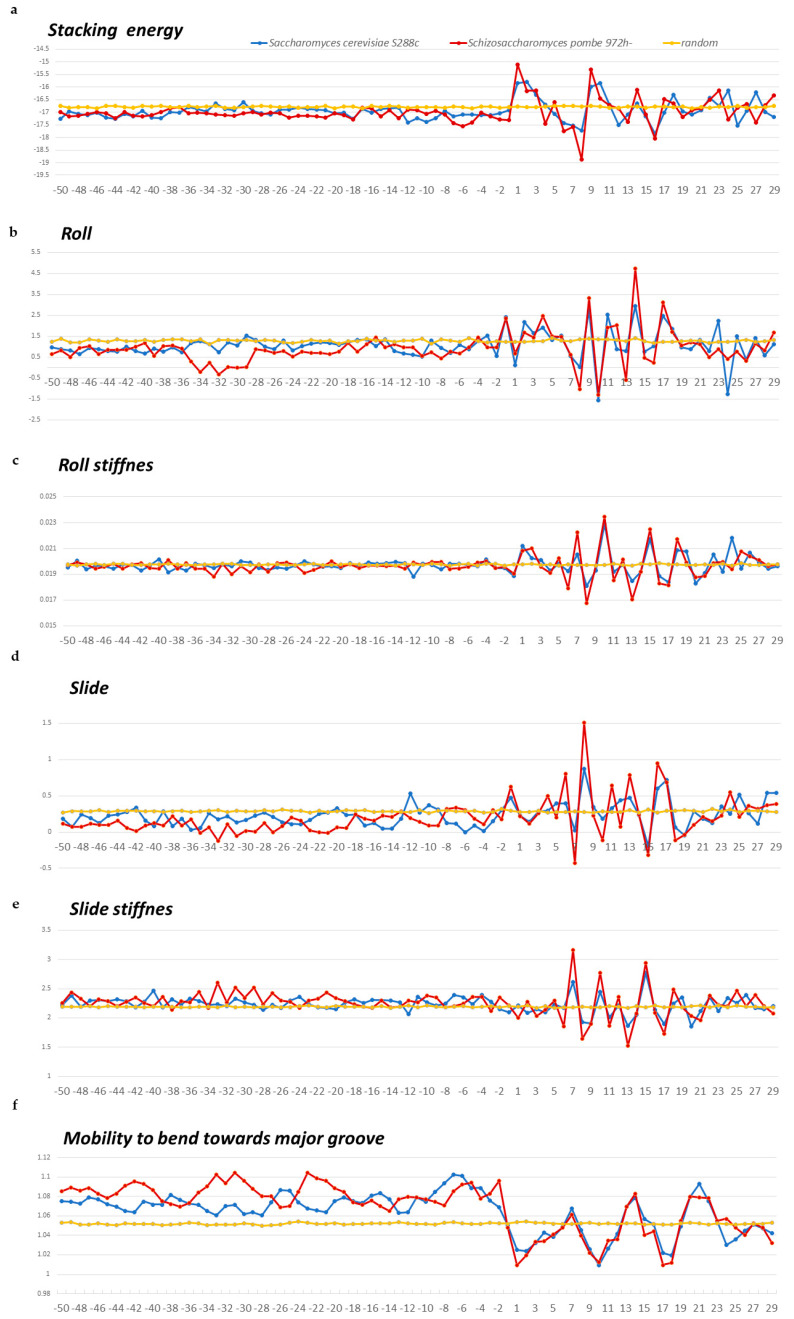
Local variations of the values of physical and structural parameters in core promoter regions of tRNA genes of *S. pombe* (in red) and *S. cerevisiae* (in blue), along with the profile of the 80 bp set of 3000 computer-simulated random nucleotide sequences (in yellow). The transcription start position is indicated by “+1” on the abscissa axis. (**a**) Stacking energy (in kcal/mol). (**b**) Roll (in degrees). (**c**) Stiffness of the duplex structure to roll alteration (in kcal/mol degree). (**d**) Slide (in angstroms). (**e**) Stiffness of the duplex structure to slide alteration (in kcal/mol angstrom). (**f**) Mobility to bend towards major groove (in mobility units).

**Figure 6 ijms-25-11758-f006:**
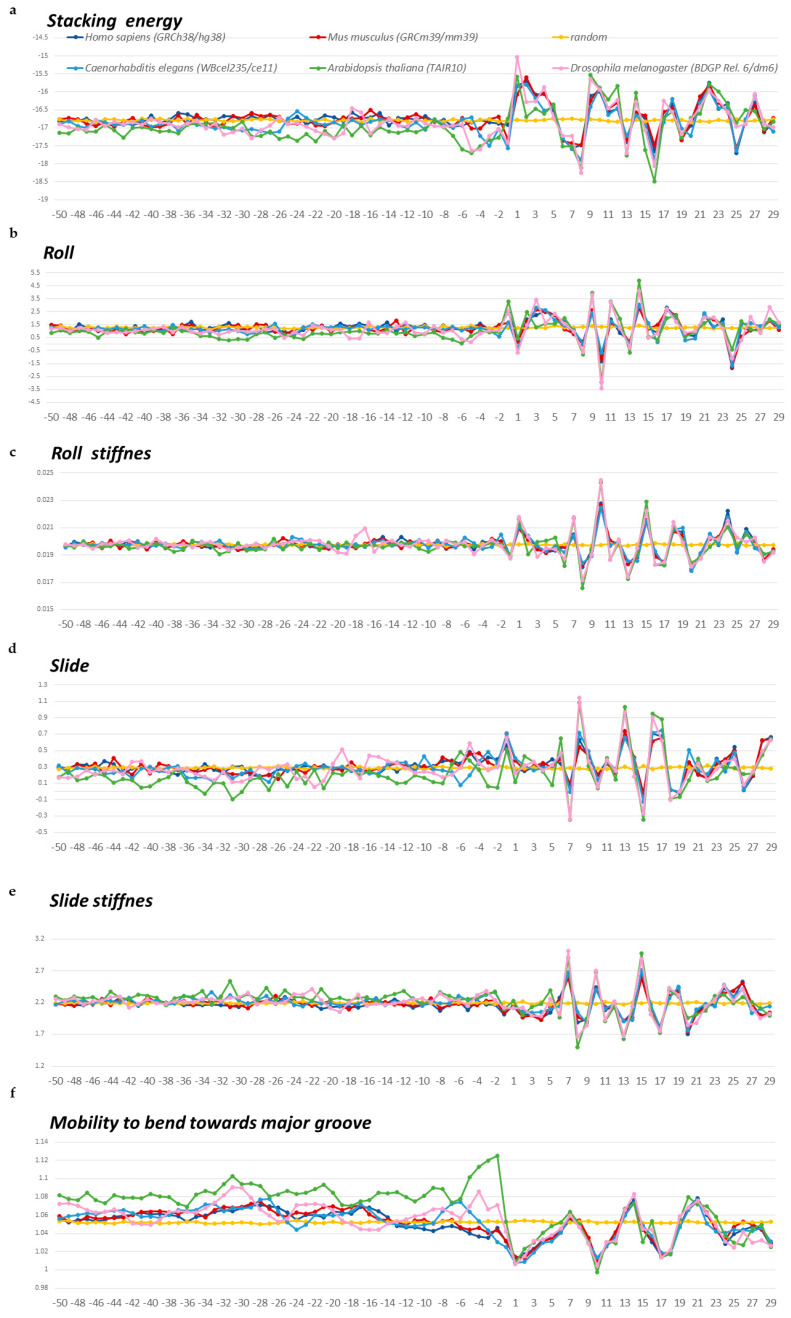
Local variations of the values of physical and structural parameters in core promoter regions of tRNA genes of *A. thaliana* (in green), *D. melanogaster* (in pink), *C. elegans* (in light blue), *M. musculus* (in red), *H. sapiens* (in blue), along with the profile of the 80 bp set of 3000 computer-simulated random nucleotide sequences (in yellow). The transcription start position is indicated by “+1” on the abscissa axis. (**a**) Stacking energy (in kcal/mol). (**b**) Roll (in degrees). (**c**) Stiffness of the duplex structure to roll alteration (in kcal/mol degree). (**d**) Slide (in angstroms). (**e**) Stiffness of the duplex structure to slide alteration (in kcal/mol angstrom). (**f**) Mobility to bend towards major groove (in mobility units).

**Figure 7 ijms-25-11758-f007:**
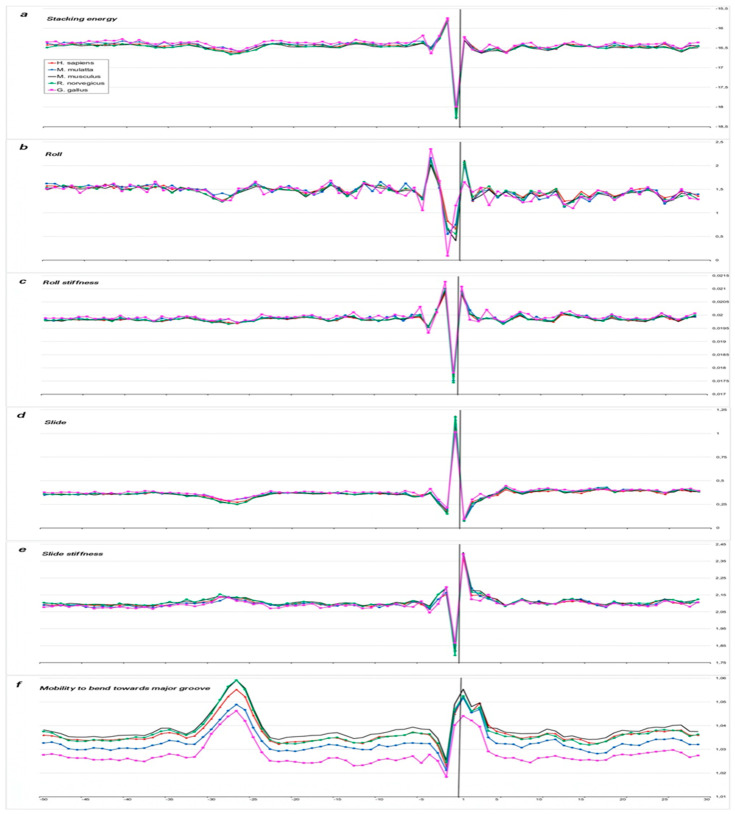
Adapted from our previous work [14]. Local variations of the values of physical and structural parameters in core promoter regions of mRNA genes transcribed by Pol II for *C. familiaris* (red), *D. melanogaster* (blue), *A. mellifera* (black), *D. rerio* (green), and *C. elegans* (pink). The transcription start position is indicated by “+1” on the abscissa axis. (**a**) Stacking energy (in kcal/mol). (**b**) Roll (in degrees). (**c**) Stiffness of the duplex structure to roll alteration (in kcal/mol degree). (**d**) Slide (in angstroms). (**e**) Stiffness of the duplex structure to slide alteration (in kcal/mol angstrom). (**f**) Mobility to bend towards major groove (in mobility units).

**Figure 8 ijms-25-11758-f008:**
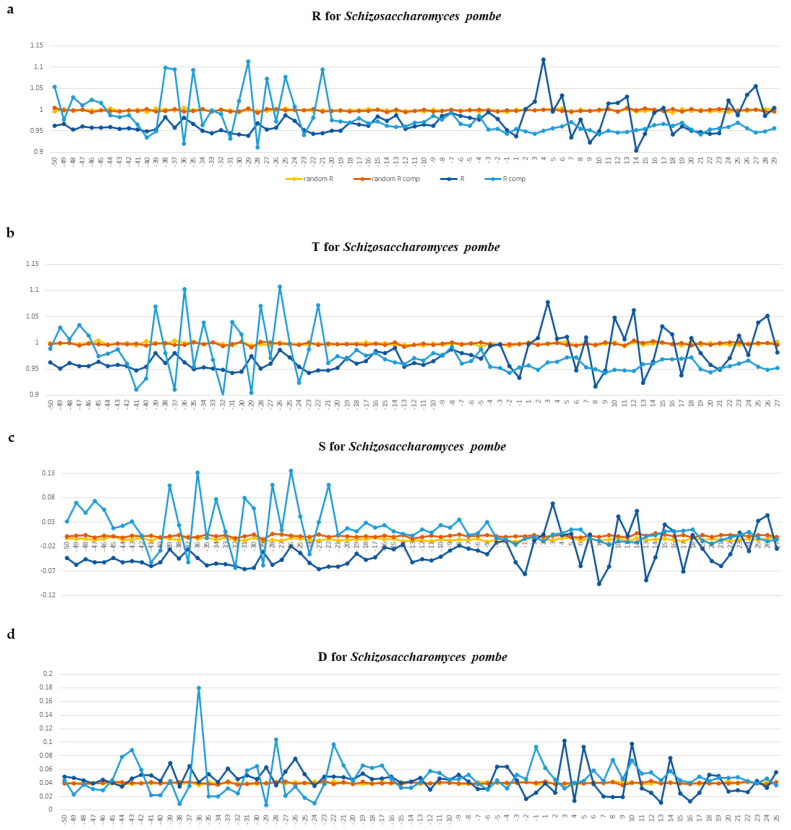
(**a**–**d**). Profiles of ultrasonic cleavage indexes and DNase I cleavage indexes for *S. pombe*. *R*—relative cleavage intensities of each of the 16 dinucleotides; upper strand is dark blue, template strand is blue. *T*—relative cleavage intensities of central position of each of 256 tetranucleotides; *S*—the combination *R* and *T*, namely, the difference between the tetranucleotide cleavage index and the dinucleotide cleavage index in its center *S* = (*T* − *R*)/*T*. If *S* < 0, the first and the fourth nucleotides of a tetranucleotide bring down the intensity of the cleavage in the central step; otherwise, they increase it. Relative cleavage intensities of each of the 16 dinucleotides (*R* index) and relative cleavage intensities of each of the 256 tetranucleotides (*T* index) were obtained from DNA fragmentation experiments by ultrasound [19]. *D*—DNase I cleavage [20]. The gene start position is indicated by “+1” on the abscissa axis.

**Figure 9 ijms-25-11758-f009:**
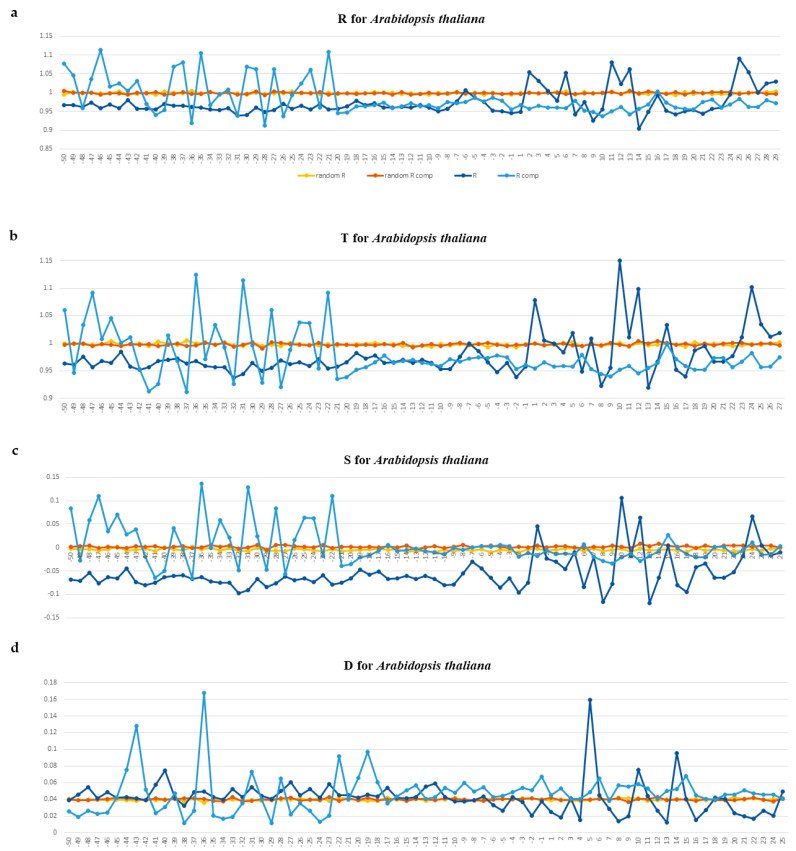
(**a**–**d**). Profiles of ultrasonic cleavage indexes and DNase I cleavage indexes for *A. thaliana*. Upper strand is dark blue, template strand is blue. *R*—relative cleavage intensities of central position of each of 16 dinucleotides; *T*—relative cleavage intensities of central position of each of 256 tetranucleotides; *S*—the combination *R* and *T*, namely, the difference between the tetranucleotide cleavage index and the dinucleotide cleavage index in its center *S* = (*T* − *R*)/*T*. If *S* < 0, the first and the fourth nucleotides of a tetranucleotide bring down the intensity of the cleavage in the central step; otherwise, they increase it. Relative cleavage intensities of each of the 16 dinucleotides (*R* index) and relative cleavage intensities of each of the 256 tetranucleotides (*T* index) were obtained from DNA fragmentation experiments by ultrasound [19]. *D*—DNase I cleavage [20]. The gene start position is indicated by “+1” on the abscissa axis.

## Data Availability

Scripts for calculations in Python and sets of sequences are available freely upon request by e-mail imb_irina@rambler.ru.

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
