# Peer review of "Structural Features of DNA in tRNA Genes and Their Upstream Sequences"

_ijms, 2024, doi:10.3390/ijms252111758_

Round 1
Reviewer 1 Report
Comments and Suggestions for Authors
The role of the protein TBP is a key factor in the transcription of eukaryotic cells. However, it is known in the field of molecular biology that the presence of a TATA-box in the eukaryotic promoters is only around 10-20%. This point is extremely relevant because the analysis presented by the author is focused in the presence of a binding-site in the promoters using genomic tools. I have the following comments to the authors:
Major Comments:
1. The analysis of the promoters in order to search for a TBP binding site is not clear because the participation of TBP as a DNA-binding factor does not occur in all eukaryotic promoters. In addition, what is the idea of searching for this element in bacterial promoters?. TBP protein is only present in eukaryotic cells.
2. The main goal of this work is to search a TATA-binding element in different promoters. However, only bioinformatics data is presented. Maybe, the title must contain this approach. In addition, to support the purpose of the work, it is necessary the addition of an experimental approach such as EMSA or ChIP to validate the genomic data. This point is crucial to support the purpose presented in the title of the work.
3. In the discussion section, there is not any comments about the distribution of the TATA-box element in eukaryotic promoters. It is possible to extract information from the EPD to generate the most accurate analysis and to identify the percentage of promoters that contain TATA-box versus TATA-less promoters and to generate a most enrichment discussion.
4. I think the discussion section is extremely poor in the analysis of the results. I think there is a lot of information such as the role of TBP in the other RNA polymerase machineries and why it is present in all. It is necessary the addition of an evolutionary analysis of the role of TBP in the expression of all kind of eukaryotic cells.
5. I do not understand why the authors added the analysis of bacterial promoters since TBP is a eukaryotic protein. This point is not adequate in the overall presentation of the work,
Author Response
Comment 1.The analysis of the promoters in order to search for a TBP binding site is not clear because the participation of TBP as a DNA-binding factor does not occur in all eukaryotic promoters. In addition, what is the idea of searching for this element in bacterial promoters?. TBP protein is only present in eukaryotic cells.
Answer 1. We have analyzed upstream region of tRNA genes in Eukaryotes and Archaea. TBP is the component of TFIIIB complex necessary for transcription by Pol III in all eukaryotes. Archaea use the orthologue of TBP. The goal of this work is to detect the positions interacting with TBP or its orthologue in the upstream region of tRNA genes in Eukaryotes and Archaea. Comparison of the DNA structure from organisms possessing TBP with organisms lacking TBP should disclose the role of spatial structure of naked DNA in regulatory regions interacting with Pol III complex proteins. That is why we have analyzed structural features of tRNA genes and their upstream regulatory regions of Bacteria, also.
- The main goal of this work is to search a TATA-binding element in different promoters. However, only bioinformatics data is presented. Maybe, the title must contain this approach. In addition, to support the purpose of the work, it is necessary the addition of an experimental approach such as EMSA or ChIP to validate the genomic data. This point is crucial to support the purpose presented in the title of the work.
Answer 2. We used both bioinformatics and experimental data on sequence features and ultrasound cleavage. We have analyzed high confidence sets from the collection of the Genomic tRNA Database (GtRNAdb) (http://gtrnadb.ucsc.edu). This database contains tRNA gene predictions made by tRNAscan-SE on complete or nearly complete genomes. These data are novel and most complete available data on tRNA gene sequences.
- In the discussion section, there is not any comments about the distribution of the TATA-box element in eukaryotic promoters. It is possible to extract information from the EPD to generate the most accurate analysis and to identify the percentage of promoters that contain TATA-box versus TATA-less promoters and to generate a most enrichment discussion.
Answer 3. We have extended the discussion section. Indeed, we have used the EPD New section of the Eukaryotic Promoter Database (EPD) (http://epd.vital-it.ch) for the analysis of mRNA promoters in eukaryotic organisms in our previous works:
Il’icheva, I. A., Khodikov, M. V., Poptsova, M. S., Nechipurenko, D. Y., Nechipurenko, Y. D., & Grokhovsky, S. L. (2016). Structural features of DNA that determine RNA polymerase II core promoter. BMC genomics, 17, 1-21.
Melikhova, A. V., Anashkina, A. A., & Il’icheva, I. A. (2022). Evolutionary Invariant of the Structure of DNA Double Helix in RNAP II Core Promoters. International Journal of Molecular Sciences, 23(18), 10873.
Savina, E. A., Shumilina, T. G., Tumanyan, V. G., Anashkina, A. A., & Il’icheva, I. A. (2023). Core promoter regions of antisense and long intergenic non-coding RNAs. International Journal of Molecular Sciences, 24(9), 8199.
- I think the discussion section is extremely poor in the analysis of the results. I think there is a lot of information such as the role of TBP in the other RNA polymerase machineries and why it is present in all. It is necessary the addition of an evolutionary analysis of the role of TBP in the expression of all kind of eukaryotic cells.
Answer 4. Thank you for this comment, we expanded and detailed the discussion section.
- I do not understand why the authors added the analysis of bacterial promoters since TBP is a eukaryotic protein. This point is not adequate in the overall presentation of the work.
Answer 5. Thanks for pointing out this problem. We have used the Bacterial profiles for the comparison of their upstream structures in the organisms possessing TBP and organisms lacking TBP.

Reviewer 2 Report
Comments and Suggestions for Authors
The review manuscript by Savina et al. describes several consensus sequences at the promoters of tRNA genes. Those results are important for researchers working on the transcriptional mechanisms of tRNA genes by pol III. Overall, the analysis is well done, and the results support the conclusions. However, I have a few observations:
1. The Discussion section could be extended to discuss which transcription factors might bind each described sequence.
2. Is there any idea how TBP binds to those tRNA promoters that do not contain a consensus TATA box?
3. A schematic figure should be included with the positions and sequence of the described motifs.
Comments on the Quality of English Language
The manuscript needs a text edition because it is full of language mistakes, and sometimes, it is hard to understand the intended meaning.
Author Response
The review manuscript by Savina et al. describes several consensus sequences at the promoters of tRNA genes. Those results are important for researchers working on the transcriptional mechanisms of tRNA genes by pol III. Overall, the analysis is well done, and the results support the conclusions. However, I have a few observations:
- The Discussion section could be extended to discuss which transcription factors might bind each described sequence.
Response 1. Thank you for this comment, we extended the discussion section. We don’t discuss separate transcription factor binding studying rather general structural properties for Pol III complex binding.
- Is there any idea how TBP binds to those tRNA promoters that do not contain a consensus TATA box?
Response 2. From the kinetic probing, it was found that TBP has less than a 10**3-fold preference for binding TATAWAAR sequence compared to binding of nonspecific yeast genomic DNA. These results allow us to suggest that hydrogen bonding does not play any role in TBP–TATA box complex formation. Therefore, those octanucleotides that are selected on the basis of low energy costs for bending towards a wide groove can be TATA elements.
- A schematic figure should be included with the positions and sequence of the described motifs
Response 3. Thank you. We have updated the figures and included new scheme (Figure 7). We described known motifs of Boxes A and B, and consensuses found.
Comments on the Quality of English Language
The manuscript needs a text edition because it is full of language mistakes, and sometimes, it is hard to understand the intended meaning.
Response 4. Thanks. We have carefully checked and revised the text.
Reviewer 3 Report
Comments and Suggestions for Authors
Review
Figure 2: Archaea and Bacteria clearly have conserved sequence around -1. For RNA processing? Please explain.
I worry that Weblogo is sensitive to changes in spacings between functional elements. Is there an AI or another computational fix for this? The segments of genes that encode tRNAs must be correlated with the gene sequences.
There must be a more sophisticated analysis that can be applied.
The pre-life tRNA sequence is known to the last nucleotide. This should be part of the current analysis.
I suggest the analysis be done on T. thermophilus and P. furiosis. These are an ancient Bacterium and an ancient Archaeon.
I suggest that a method be found (i.e., an improved logo technique or artificial intelligence method) to improve the way Weblogo deals with different spacings in sequences being compared to see whether stronger logos might be obtained.
A major criticism:
The text does not relate Box A (D loop) and Box B (T stem-loop-stem) to the highly conserved tRNA RNA sequence. Whether or not this has previously been shown, it should be emphasized here.
Box A:
Pre-life tRNA sequence:
D loop
1-GCGGCGGUAGCCUAGCCUAGCCUAGGCGGCCGGG-34
TRGYNNAR--BGG-+8/+18
Box B
GGUUCRANYCY
CCGGGUUCAAAUCCCGG T stem-loop-stem
(((((*******)))))
See:
https://www.preprints.org/manuscript/202407.0578/v1
https://www.mdpi.com/2075-1729/13/11/2224
Much of the analyses shown are not highly informative and strong conclusions are not reached by the authors.
Overall, the analyses do not appear to lead to strong or reliable conclusions. I suggest that additional studies be done prior to publication.
Comments on the Quality of English Language
The paper is mostly readable with a few grammatical errors. The writing of the paper should be improved. The authors should avoid 1 sentence paragraphs. The authors should not use "since" to mean "because".
Author Response
Comments 1. Figure 2: Archaea and Bacteria clearly have conserved sequence around -1. For RNA processing? Please explain.
Response 1. Thanks for the comment. This problem was discussed previously. The sequence around -1 is conserved as in upstream regions of tRNA, so in mRNA promoters.
This position is responsible for the double helix divergence, so the dinucleotide step that it occupies must have unique properties. It is known that the deformability of dinucleotides decreases in the order of PyPu > PuPu > PuPy. It was shown that with the help of a spin probe while studying the effects of nucleotide sequence on DNA duplex dynamics. The special mobility of PyPu steps is explained by the greater intensity of the S=>N dynamics in furanose cycles in 5’-terminal pyrimidines compared to 5’-terminal purines, and after 5’Cyt, it reaches its maximum. The advantage of the CpA step over CpG in positions -1 and +1 can be explained by the presence of only two hydrogen bonds, which must be broken at the initial stage of chain divergence. This explanation is confirmed by reactivity with the conformation-sensitive reagent chloroacetaldehyde, which reacts with unpaired adenines and cytosines. This reactivity was confined strictly to adenosine in the d(CA/TG) repeat.
Comments 2. I worry that Weblogo is sensitive to changes in spacings between functional elements. Is there an AI or another computational fix for this? The segments of genes that encode tRNAs must be correlated with the gene sequences.
There must be a more sophisticated analysis that can be applied.
The pre-life tRNA sequence is known to the last nucleotide. This should be part of the current analysis.
Response 2. Thanks for the comments. We discussed role of conservative sequences in forming of special structure of mature tRNA. We refer to relevant publication on this problem.
We used phased sequences of tRNA gene promoters. The sequences used were rather short, namely 80 and 60 bp. There are no known binding sites, and no spacing to study. We used most relevant database on promoters of genes coded by Pol III , and use complete set of structural and physical chemical characteristics to study pattern.
Yes, WebLogo is sensitive to alignment. We studied structural patterns of promoter regions. Spacing between nucleotides affects only logo, but not structural properties.
Comments 3. I suggest the analysis be done on T. thermophilus and P. furiosis. These are an ancient Bacterium and an ancient Archaeon.
I suggest that a method be found (i.e., an improved logo technique or artificial intelligence method) to improve the way Weblogo deals with different spacings in sequences being compared to see whether stronger logos might be obtained.
Response 3. Thanks for the suggestion. We used more reliable data on representative organisms from different domains of life. Analysis of the sequences from the representative organisms didn’t reveal the difference between Bacteria studied. We believe this work is complete. We may use T. thermophilus and P. furiosis data in our next work. Thanks again for the suggestion.
As we pointed out above, we studied structural properties of phased sequences. Alignment is an additional tool to find sequence patterns. It is out of the frames of the current study.
Comments 4. A major criticism:
The text does not relate Box A (D loop) and Box B (T stem-loop-stem) to the highly conserved tRNA RNA sequence. Whether or not this has previously been shown, it should be emphasized here.
Box A:
Pre-life tRNA sequence:
D loop
1-GCGGCGGUAGCCUAGCCUAGCCUAGGCGGCCGGG-34
TRGYNNAR--BGG-+8/+18
Box B
GGUUCRANYCY
CCGGGUUCAAAUCCCGG T stem-loop-stem
(((((*******)))))
See:
https://www.preprints.org/manuscript/202407.0578/v1
https://www.mdpi.com/2075-1729/13/11/2224
Much of the analyses shown are not highly informative and strong conclusions are not reached by the authors.
Overall, the analyses do not appear to lead to strong or reliable conclusions. I suggest that additional studies be done prior to publication.
Response 4. Thanks for the comments. We discussed Box A (D loop) and Box B (T stem-loop-stem) in detail, and show these sequences. We found structural properties of DNA preferable for these sequence location in promoters.
We extended the text and rearranged the figures correspondingly.
We cited the suggested reference as well as other relevant paper to clarify the study.
The conclusion and results section were rewritten.
Comments on the Quality of English Language
Comments 5. The paper is mostly readable with a few grammatical errors. The writing of the paper should be improved. The authors should avoid 1 sentence paragraphs. The authors should not use "since" to mean "because".
Response 5. Thanks for the comments. We checked English presentation and have rewritten the text. We checked grammar style as well.
Reviewer 4 Report
Comments and Suggestions for Authors
In this manuscript, the authors used computational methods to describe the nucleotide and positional conservation of promoter elements (TATA, A, and B-boxes) of tRNAs across species of multiple domains. They further included DNase- and ultrasonic-sensitivity to model structural features. The findings are numerous and described well within the paper, and I believe the species represented in each group to be suitable. The analyses should be of interest to those in the field of tRNA or mechanisms of Pol III transcription.
Minor text edits:
Page 2 line 6: remove the apostrophe in it's
Page 15 line 26: change find to found
Author Response
Comment 1: In this manuscript, the authors used computational methods to describe the nucleotide and positional conservation of promoter elements (TATA, A, and B-boxes) of tRNAs across species of multiple domains. They further included DNase- and ultrasonic-sensitivity to model structural features. The findings are numerous and described well within the paper, and I believe the species represented in each group to be suitable. The analyses should be of interest to those in the field of tRNA or mechanisms of Pol III transcription.
Minor text edits:
Page 2 line 6: remove the apostrophe in it's
Page 15 line 26: change find to found
Response 1: Authors thanks the reviewer for careful reading and valuable comments. Minor edits done.
Round 2
Reviewer 1 Report
Comments and Suggestions for Authors
The new version of the manuacript is better than the previous version. The responses of the authiors to my comments are adequated. I think that this version of the manuscript is able to be published.
Author Response
Comment 1: The new version of the manuacript is better than the previous version. The responses of the authiors to my comments are adequated. I think that this version of the manuscript is able to be published.
Response 1: Authors thank the reviewer for careful reading and valuable comments.
Reviewer 3 Report
Comments and Suggestions for Authors
This paper requires significant re-writing and re-thinking. The paper is very challenging to read because of the poor English style and presentation. The paper is poorly organized and presented. The paper presents a lot of data without much clarity about what the data may mean.
The TSS (transcription start site) is not clearly identified in figures. Is there confusion about the 5’-end of the tRNA versus the TSS? I believe that tRNAs are generated by 5’- and 3’-RNA processing. For instance, eukaryotic tRNAs have a unique Pol III 5’-CAP structure that must be removed to form a functional tRNA. Bacterial tRNA genes are often embedded in operons with ribosomal genes. These tRNA genes do not (so far as this reviewer knows) have tRNA-specific promoters. If there are bacterial tRNA-specific promoters, I am not aware of these. The authors must explain. They appear to be searching for promoters in the wrong places without sorting different tRNA genes that may be organized differently.
Eukaryotic tRNA genes have dual purposes: 1) tRNA encoding; and 2) internal tRNA promoters (Box A and Box B). For clarity of presentation, the authors must make this distinction clear from the start of the paper. There is a small gesture to this issue in Discussion, but the data in the paper is not easy to understand without this explanation early.
Box A is clearly derived from the tRNA D loop. Box B is clearly derived from the tRNA T stem-loop-stem. More should be made of these insights. I have never before known this about Pol III transcription. Clearly, evolution of these sequences must support tRNA function above promoter function but must not critically impair promoter function.
It is this reviewer’s strong opinion that this paper must be reorganized and re-thought before publication.
Comments on the Quality of English Language
The manuscript needs review by a native English writer. It is very difficult to read.
Author Response
Many thanks to the reviewer for his careful reading of the manuscript and substantial comments. They allowed us to rethink some results and substantially revise the text.
Please, read below answers for all your comments.
Note 1: The TSS (transcription start site) is not clearly identified in figures.
Answer 1: The transcription start position is indicated by “+1” on the abscissa axis. Appropriate explanation added to the figures’ captions.
Note 2: Is there confusion about the 5’-end of the tRNA versus the TSS? I believe that tRNAs are generated by 5’- and 3’-RNA processing. For instance, eukaryotic tRNAs have a unique Pol III 5’-CAP structure that must be removed to form a functional tRNA. Bacterial tRNA genes are often embedded in operons with ribosomal genes. These tRNA genes do not (so far as this reviewer knows) have tRNA-specific promoters. If there are bacterial tRNA-specific promoters, I am not aware of these. The authors must explain. They appear to be searching for promoters in the wrong places without sorting different tRNA genes that may be organized differently.
Answer 2: We used complete genomic tRNA samples ("high confidence" sets, so they are not the matured tRNA) from the GtRNAdb database. This database obtained by analyzing whole genome sequences using the tRNAscan-SE 2.0 application [Chan, P. P., Lin, B. Y., Mak, A. J., & Lowe, T. M. (2021). tRNAscan-SE 2.0: improved detection and functional classification of transfer RNA genes. Nucleic acids research, 49(16), 9077-9096], reference [30] in the text. The annotation of this application states: «tRNA-derived repetitive elements, whose primary sequences are very similar to real tRNA genes, have been commonly found in a lot of vertebrates, some worms, and some plants. To address this problem, we applied a multi-step post-filtering process to the predictions in large eukaryotes by using EukHighConfidenceFilter from tRNAscan-SE 2.0. The tool assesses the predictions with a combination of domain-specific, isotype-specific, and secondary structure scores in two filtering stages on top of the pseudogene classification, and determines the "high confidence" set of genes that are most likely to be functioned in the translation process. A small number of the predictions that have high scores but atypical features such as unexpected anticodons are separately marked for further investigation.». As for bacterial tRNA genes, which indeed often integrated into operons with ribosomal genes, however, it does not matter to us in the case of study. 5’-flanking regions of tRNA genes of bacteria were used exclusively for comparing the features of their spatial structure characteristics with the 5’-flanking regions of tRNA genes of eukaryotes and archaea. In addition, we have added references to studies that have examined promoters of E. coli in the text.
Note 3. Eukaryotic tRNA genes have dual purposes: 1) tRNA encoding; and 2) internal tRNA promoters (Box A and Box B). For clarity of presentation, the authors must make this distinction clear from the start of the paper. There is a small gesture to this issue in Discussion, but the data in the paper is not easy to understand without this explanation early.
Answer 3. We have taken this request into account and at the beginning of the text we pointed out these features of the organization of tRNA gene promoters.
Note 4. Box A is clearly derived from the tRNA D loop. Box B is clearly derived from the tRNA T stem-loop-stem. More should be made of these insights. I have never before known this about Pol III transcription. Clearly, evolution of these sequences must support tRNA function above promoter function but must not critically impair promoter function.
Answer 4. Thank you for this valuable comment. We have added a discussion of the relationship between the main promoters, boxes A and B, and the structural features of mature tRNA to the text, and have also touched on the evolutionary aspect of this problem.
Note. 5. The manuscript needs review by a native English writer.
Answer 5. We asked a native English speaker to review the manuscript.
Round 3
Reviewer 3 Report
Comments and Suggestions for Authors
According to the authors: "The aim of this work is to detect the possible position for TBP binding in the 5’-flanking regions of tRNA genes in Eukaryotes and Archaea."
Is this really the aim of this work?
The paper appears to be too preliminary for publication in its current form. The paper includes too many errors.
The abstract is poorly written
Can the authors use AI or an English editor to improve their English?
The paper is difficult to read
The authors mix up the 5’ end of the tRNA coding gene and the transcription start site. See:
Genetics. 2013 May; 194(1): 43–67. doi: 10.1534/genetics.112.147470
PMCID: PMC3632480PMID: 23633143
Transfer RNA Post-Transcriptional Processing, Turnover, and Subcellular Dynamics in the Yeast Saccharomyces cerevisiae
Anita K. Hopper
The current paper does not come to strong or reliable conclusions
Much more must be known about TFIIIC and TFIIIB interaction with tRNA promoters than is reviewed or utilized in this paper.
That BoxA is derived from the tRNA D loop is interesting.
That BoxB is derived from the tRNA T loop is interesting.
If this observation is novel, why isn’t this observation highlighted in the paper? It is buried in the Discussion.
This paper requires significant improvement for publication.
Comments on the Quality of English Language
The English writing is very poor. Please get help from an English editor and/or utilize AI support or Grammarly.
Author Response
Comments and Suggestions for Authors
1) According to the authors: "The aim of this work is to detect the possible position for TBP binding in the 5’-flanking regions of tRNA genes in Eukaryotes and Archaea."
Is this really the aim of this work?
Response: Thank you for this question. Yes, the purpose of this paper is to discover a possible position for TBP binding, and that is what prompted the study. But the overall goal of this work was broader, namely. The aim of this work is to identify the positions of intragenic and extragenic Pol III promoters that regulate transcription of tRNA genes in eukaryotes and archaea.Specifically, to identify the presence of possible TBP binding sites in the 5'-flanking regions of tRNA genes in eukaryotic and archaea organisms. We compare them with 5'-flanking regions of tRNA genes in bacteria, as well as with Pol II promoters in eukaryotes, and identify features of mechanical and structural properties of double-stranded DNA in promoter regions of different genes.
2) The paper appears to be too preliminary for publication in its current form. The paper includes too many errors.
Response: What errors did you find in the manuscript? We would be grateful if you could point out specific errors.
3) The abstract is poorly written. Can the authors use AI or an English editor to improve their English? The paper is difficult to read
Response: We appreciate this comment. We have improved the language both in the abstract and in the text of the manuscript.
4) The authors mix up the 5’ end of the tRNA coding gene and the transcription start site. See:
Genetics. 2013 May; 194(1): 43–67. doi: 10.1534/genetics.112.147470
PMCID: PMC3632480PMID: 23633143
Transfer RNA Post-Transcriptional Processing, Turnover, and Subcellular Dynamics in the Yeast Saccharomyces cerevisiae_Anita K. Hopper
Response: We are grateful to the reviewer for information about an interesting review of experimental studies of post-transcriptional changes in the primary tRNA transcript in S. cerevisiae. However, the reviewer does not seem to have correctly assessed the subject we are studying. We are studying genomic sequences, namely the sequences of tRNA genes and their 5'-flanking fragments. Transcription starts at position +1, and a TSS site is commonly referred to as the nearest neighborhood of positions -1/+1. In this position, in the overwhelming number of sequences there is a Pyr/Pu staple, which facilitates chain divergence for the transition to the elongation stage. A similar situation occurs at the transcription start sites of mRNA transcribed by the Pol II apparatus. The reason why the -1/+1 positions most often contain the Pyr/Pu staple is that it is this dinucleotide variant that facilitates the DNA strand divergence necessary for transcription. The conformational features of Pyr/Pu dinucleotide have been described in detail in our previous studies.
[13] Il’icheva, I. A., Khodikov, M. V., Poptsova, M. S., Nechipurenko, D. Y., Nechipurenko, Y. D., &Grokhovsky, S. L. (2016). Structural features of DNA that determine RNA polymerase II core promoter. BMC genomics, 17, 1-21.
[14] Melikhova, A. V., Anashkina, A. A., &Il’icheva, I. A. (2022). Evolutionary Invariant of the Structure of DNA Double Helix in RNAP II Core Promoters. International Journal of Molecular Sciences, 23(18), 10873.
The transcription start position is also clearly visible in all profiles of DNA physical properties and in the profiles of DNA cleavage intensity indices by ultrasound and DNase I, since the properties of DNA structure change markedly during the transition from the flanking site to the gene site.
Here is an example of one of the analyzed sequences from S. cerevisiae tRNA genes, and an image of the structure of a mature tRNA having this sequence
tRNA-Ala-AGC-1-1 chrIV:410379-410451 (+)
Upstream / Downstream Sequence
GTAACGACCATACAAATATT / AATTATTTTTTACTTTCCGC
Saccharomyces_cerevisiae_tRNA-Ala-AGC-1-1 (tRNAscan-SE ID: chrIV.trna3) Ala (AGC) 73 bp Sc: 69.6 chrIV:410379-410451 (+)
GTAACGACCATACAAATATTGGGCGTGTGGCGTAGTCGGTAGCGCGCTCCCTTAGCATGGGAGAGGTCTCCGGTTCGATTCCGGACTCGTCCAAATTATTTTTTACTTTCCGC
5) The current paper does not come to strong or reliable conclusions
Response: We cannot agree with this statement of the reviewer.
First, convincing evidence has been obtained that the properties of the 5'-flanking regions of tRNA genes, which are transcribed by Pol III, differ significantly from the promoter regions of mRNA genes, as well as promoters of antisense and long noncoding intergenic RNAs transcribed by Pol II. The main difference between the 5'-flanking regions of tRNA genes (in which extragenic eukaryotic and archaea promoters may be present) is that they do not contain a singular region near the -28 bp position, which is the TBP receptor in mRNA promoters. In contrast to Pol II promoters, the 5'-flanking regions of tRNA genes contain several fragments of 8 bp in length that are capable of anchoring TBP, but the affinity of these fragments for TBP is significantly lower than that of the TATA-box mRNA.
We identified specific properties in a fragment located immediately upstream of the TSS from position 18±1 before transcription initiation. We hypothesize that this 20-bp fragment contacts TFIII3C and, possibly, simultaneously TFIIID, thus determining the position of their interaction.
Bacteria in the 5'-flanking regions of tRNA genes do not contain singular regions; however, the physical properties of bacterial DNA in these regions reveal the possibility of interaction with proteins that have affinity for the minor groove of DNA.
Another obvious result of the work, obtained for the first time, is the statement that organisms of all three domains contain two undecanucleotides (Boxes A and B), which are located quite identically with respect to the beginning (Box A) and the end (Box B) of genes in eukaryotes and archaea, while in bacteria Box B is shifted only two positions to the center. The consensus sequences of each of these boxes are identical in all organisms, all three domains. The differences concern only the information content of a number of positions. The constancy of the location of the boxes indicates that they participate in the formation of the D-loop and TψC-loop of mature tRNAs.
6) Much more must be known about TFIIIC and TFIIIB interaction with tRNA promoters than is reviewed or utilized in this paper.
Response: Our data indicate a complex multistep regulation of tRNA transcription by the Pol III apparatus. Features of the 5'-flanking regions indicate that the TBP protein, which is a fragment of the TFIIIB complex, is not a determinant regulator of transcription.
This result is fully consistent with recent experimental studies of Pol III work to which we refer:
[28] Talyzina, A.; Han, Y.; Banerjee, C.; Fishbain, S.; Reyes, A.; Vafabakhsh, R.; He, Y. Structural Basis of TFIIIC-Dependent RNA Polymerase III Transcription Initiation. Mol. Cell2023, 83, 2641-2652.e7, doi:10.1016/j.molcel.2023.06.015.and
[29] van Breugel, M. E., Gerber, A., & van Leeuwen, F. (2024). The choreography of chromatin in RNA polymerase III regulation. Biochemical Society Transactions, 52(3), 1173-1189.
7) That Box A is derived from the tRNA D loop is interesting. That Box B is derived from the tRNA T loop is interesting.
If this observation is novel, why isn’t this observation highlighted in the paper? It is buried in the Discussion.
Response: Thank you. We highlighted this result immediately after Logo-profiles, as well in the Discussion, and Conclusion.
8) This paper requires significant improvement for publication.
Comments on the Quality of English Language
9) The English writing is very poor. Please get help from an English editor and/or utilize AI support or Grammarly.
Response: Thank you. The work has been thoroughly rewritten by an English editor

Round 4
Reviewer 3 Report
Comments and Suggestions for Authors
This is the fourth submission of this paper.
What the authors identify as the transcription start site is not the transcription start site. It is the first base of the tRNA. The transcription start site is upstream. RNaseP processes to the 5'-acceptor stem of the tRNA. Why does this error in the manuscript persist?
Box A and Box B are tRNA coding regions. Differences in these regions are directed by tRNA evolution in different organisms.